METHODS

# Robust expansion of phylogeny for fast-growing genome sequence data

Yongtao Ye[1,2], Marcus H. Shum[1,2], Joseph L. Tsui[1,2], Guangchuang Yu[3], David K. Smith[1], Huachen Zhu[1,2,4,5], Joseph T. Wu[1,2], Yi Guan[1,2,4,5], Tommy Tsan-Yuk Lam[1,2,4,5,6] *

1 State Key Laboratory of Emerging Infectious Diseases, School of Public Health, The University of Hong Kong, Hong Kong SAR, P. R. China, 2 Laboratory of Data Discovery for Health Limited, 19W Hong Kong Science & Technology Parks, Hong Kong SAR, P. R. China, 3 Department of Bioinformatics, School of Basic Medical Sciences, Southern Medical University, Guangzhou, Guangdong, China, 4 Guangdong-Hongkong Joint Laboratory of Emerging Infectious Diseases, Joint Institute of Virology (Shantou University/The University of Hong Kong), Shantou, Guangdong, P. R. China, 5 EKIH (Gewuzhikang) Pathogen Research Institute, Futian District, Shenzhen City, Guangdong, P. R. China, 6 Centre for Immunology & Infection Limited, 17W Hong Kong Science & Technology Parks, Hong Kong SAR, P. R. China

☯ These authors contributed equally to this work.
* ttylam@hku.hk

**Data Availability Statement:** The benchmark datasets and source codes are available at https://github.com/id-bioinfo/TIPars. SARS2-CoV-2 data used in this work were all downloaded from GISAID (https://www.gisaid.org/) of which

## Abstract

Massive sequencing of SARS-CoV-2 genomes has urged novel methods that employ existing phylogenies to add new samples efficiently instead of *de novo* inference. 'TIPars' was developed for such challenge integrating parsimony analysis with pre-computed ancestral sequences. It took about 21 seconds to insert 100 SARS-CoV-2 genomes into a 100k-taxa reference tree using 1.4 gigabytes. Benchmarking on four datasets, TIPars achieved the highest accuracy for phylogenies of moderately similar sequences. For highly similar and divergent scenarios, fully parsimony-based and likelihood-based phylogenetic placement methods performed the best respectively while TIPars was the second best. TIPars accomplished efficient and accurate expansion of phylogenies of both similar and divergent sequences, which would have broad biological applications beyond SARS-CoV-2. TIPars is accessible from https://tipars.hku.hk/ and source codes are available at https://github.com/id-bioinfo/TIPars.

## Author summary

Since the beginning of the COVID-19 pandemic, over 15 million SARS-CoV-2 genome sequences have been made publicly available. As sequencing cost decreases, the rate of genome sequencing is expected to greatly increase in the future and will generate numerous sequences where conventional *de novo* phylogenetic inference may no longer be suitable. TIPars allows rapid and memory-efficient expansion of phylogeny at high accuracy. This enables real-time monitoring of pathogen transmission during a pandemic using large-scale global phylogenetic analysis such as the ever-increasing SARS-CoV-2 genome sequences. We believe that the development of next-generation phylogenetic methods is

Accession Ids are shown in the directories SARS2-100k and SARS2-660k under https://github.com/id-bioinfo/TIPars/tree/master/Benchmark_datasets/.

**Funding:** This project is supported by the National Natural Science Foundation of China's Excellent Young Scientists Fund (Hong Kong and Macau) (31922087; TL), the Hong Kong Research Grants Council's General Research Fund (17150816; TL), the Health and Medical Research Fund (COVID1903011-WP1; TL), the Innovation and Technology Commission's InnoHK funding (D24H; TL,JW,YG,HZ), and the Guangdong Government for the funding supports (2019B121205009, HZQB-KCZYZ-2021014, 200109155890863, 190830095586328 and 190824215544727; YG, HZ). The funders had no role in study design, data collection and analysis, decision to publish, or preparation of the manuscript.

**Competing interests:** The authors have declared that no competing interests exist.

imperative for analysing enormous, fast-growing genome sequence datasets to gain critical evolutionary insights that, as evident in this pandemic, have real-world applications.

This is a *PLOS Computational Biology* Methods paper.

## Introduction

Next-generation sequencing (NGS) technology enables large-scale exploration of the diversity and monitoring temporal evolution of organisms, which often involve generating and analysing large numbers of sequences from new organisms on an ongoing basis. For instance, more than 15 million SARS-CoV-2 genomes have been sequenced within two years of the pandemic [1], crucial in transmission tracking and disease control. Conventional methods of *de novo* phylogeny inference, such as those implemented in IQ-TREE2 [2] and FastTree2 [3], that build the tree from scratch after collecting all relevant sequences, are unsuitable for such rapidly growing huge sequence datasets. Therefore, placing new sequences directly in existing reference trees becomes a more efficient alternative. Such 'phylogenetic placement' has been useful for taxonomic classification, while accumulative addition of sequences (incrementing the phylogeny as a result) allows efficient update of the growing phylogeny in a global context.

Previously published methods such as PhyClass [4], EPA-ng [5] and pplacer [6] use minimum evolution or maximum likelihood criteria to infer the evolutionary position of a query sequence and place it directly onto a pre-built phylogeny. These algorithms require relatively large computer memory or long runtimes that make massive sequence insertion difficult. Recently, UShER [7] was developed to tackle this problem in tracking the diversity of the numerous of SARS-CoV-2 virus genomes. The program calculates the 'branch parsimony score' to search for positions of taxa placement based solely on sequence mutations to a particular reference, and performs substantially faster than other existing programs. Although UShER showed promising placement accuracy on the SARS-CoV-2 genomes, its performance for more divergent genome sequences has not been studied.

We hereby introduce a new approach, TIPars, that inserts sequences into a reference phylogeny based on parsimony criteria aided by a full multiple sequence alignment of taxa and pre-computed ancestral sequences. Ancestral sequences are useful in assisting the efficient search of the best placement position as they often contain rich information in the evolutionary context of a phylogenetic tree [8]. Recent ancestral sequence reconstruction methods such as PastML [9] and RASP4 [10] have improved speed and accuracy to become feasible for the huge SARS-CoV-2 phylogeny. TIPars finds the position for insertion by calculating the triplet-based minimal substitution score for the query sequence on all branches (Fig 1A). To compare the performance of 8 phylogenetic placement/insertion methods, TIPars, UShER v0.3.8, EPA-ng v0.3.8, APPLES-2 v2.0.9 [11], IQ-TREE2 v2.1.3, RAPPAS v1.21 [12], PAGAN2 v.1.53 [8] and MAPLE v0.3.4 [13], we run them on four benchmark datasets (SARS-CoV-2, Influenza virus, Newcastle disease virus and 16S rRNA genes). We first tested single taxon placement by pruning one taxon from a given phylogenetic tree and applying the methods to place it back. The second test was multiple taxa insertion in which a set of taxa was removed and sequentially inserted back. We compared the topologies and log-likelihoods of the trees before pruning and after reinsertion. Our evaluation aimed to assess the robustness of the methods on both highly similar and divergent sequences, and whether the phylogenetic tree could be efficiently updated with continuously generated new sequences.

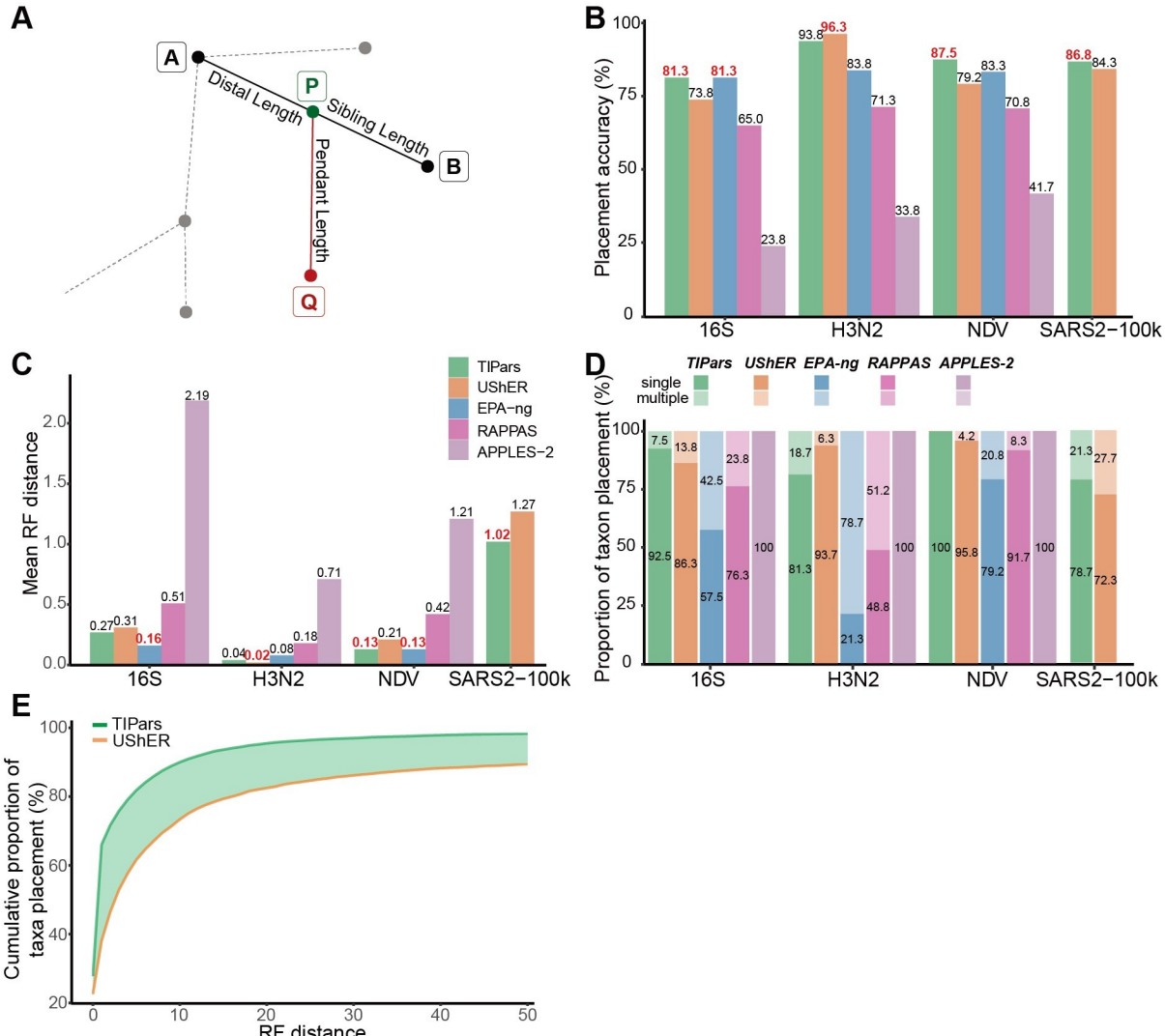

**Fig 1. Phylogenetic placement and single taxon placement performance.** (A) Illustration of the placement for a query sequence. "Q" indicates the query sequence, and "A" and "B" represent existing nodes in the reference tree. "P" represents the parental node of "Q" generated by TIPars. The minimum substitution score was calculated based on the triplet formed by A-B-Q. (B) Bar charts representing the accuracy of single taxon placement on 16S, H3N2, NDV and SARS2-100k (SARS2) datasets using TIPars (green), UShER (orange), EPA-ng (blue), RAPPAS (pink) and APPLES-2 (purple) respectively. Accuracy is indicated at the top of each bar. The highest accuracy in each dataset is highlighted in red. A true positive placement was defined by a zero RF distance between the resulting tree (after a query taxon insertion) and the reference tree. (C) Bar charts representing the mean RF distance calculated from the single taxon placement results on 16S, H3N2, NDV and SARS2-100k (SARS2) datasets using TIPars (green), UShER (orange), EPA-ng (blue), RAPPAS (pink) and APPLES-2 (purple) respectively. Mean RF distance is indicated at the top of each bar. The lowest mean RF distance in each dataset is highlighted in red. Panel (B) and (C) share the same figure legend in (C). (D) Stacked bar charts showing the proportions of single and multiple placement results for TIPars, UShER, EPA-ng, RAPPAS and APPELS-2 on 16S, H3N2, NDV and SARS2-100k datasets. Proportions with > 0.1% are indicated within the bars. TIPars generated 23% fewer multiple placements than UShER in SARS2-100k dataset. (E) Cumulative proportions of single taxon placement results on the SARS2-660k dataset at different RF distance cut-offs. The highlighted area represents the difference between TIPars and UShER.

## Results

### Computational performance of TIPars and other methods

A number of approaches have been proposed for phylogenetic placement or insertion but are impractical or computationally prohibitive for dealing with the vast number of SARS-CoV-2

**Table 1. Average running time and memory used for inserting/placing 100 genome samples into SARS2-100k reference tree.**

| Tools | CPU cores assigned | Mean insertion time (HH:MM:SS) | Mean total runtime (HH:MM:SS) | Mean peak memory (GB) |
|---|---|---|---|---|
| TIPars | 64 | 0:00:21 | 0:00:52 | 1.39 |
| TIPars | 8 | 0:00:31 | 0:01:03 | 1.18 |
| UShER | 64 | 0:00:02 | 0:03:14 | 0.84 |
| UShER | 8 | 0:00:05 | 0:05:14 | 0.16 |
| EPA-ng | 64 | 0:04:45 | 0:10:25 | 1022.14 |
| APPLES-2 | 64 | N/A | 0:04:25 | 8.38 |
| IQ-TREE2 | 64 | N/A | 5:49:10 | 101.10 |
| RAPPAS | 64 | N/A | N/A | N/A |
| PAGAN2 | 64 | N/A | N/A | N/A |
| MAPLE | 1 | 0:00:33 | 0:03:58 | 24.04 |

Tests were run on a server with 64 Intel Xeon Gold 6242 CPU cores and 1500 GB RAM for 10 repeated runs. We also compared TIPars with UShER on a general computer with 8 CPU cores. TIPars ran with JAVA setting -Xmx1G. "Insertion time" indicates the time only for the key step of searching the best position to place or insert the query while "total runtime" counts the total time running the program. RAPPAS and PAGAN2 could not be done within 96 hours, no data here. APPLES-2 and IQ-TREE2 do not report insertion time. MAPLE is a single-thread program indicating by 1 CPU core assigned. N/A indicates that data are not applicable. Total runtimes for different programs consist of 1) TIPars: "insertion time" (IT) + I/O processing (I/O); 2) UShER: IT + I/O + 'mutation-annotated tree' computation; 3) EPA-ng: IT + I/O + likelihood weight ratio (LWR) computation; 4) RAPPAS: IT + I/O; 5) APPLES-2, IQTREE2 and PAGAN2: their total running times and 6) MAPLE: IT+I/O+'substitution matrix' (without topology refinement).

genome sequences. We generated a reference SARS-CoV-2 phylogenetic tree (SARS2-100k) from 96,020 unmasked, high-quality SARS-CoV-2 sequences (detailed in Methods), and evaluated our program alongside UShER, EPA-ng, APPLES-2, IQ-TREE2, RAPPAS, PAGAN2 and MAPLE by sequentially inserting 100 new sequences. Table 1 presents the summary of runtimes for running the whole program as well as the placement/insertion process that is the key step of searching the best position to place or insert the 100 queries into the reference tree. Placement/insertion time is not available for the tools APPLES-2, IQ-TREE2, RAPPAS and PAGAN2. Only TIPars, UShER and MAPLE were practicable in terms of insertion time and memory usage. RAPPAS and PAGAN2 were unable to complete within 96 hours, hence, no data were available. Although IQ-TREE2 required the longest runtime among all programs, it had a lower peak memory than EPA-ng, which used about 1 terabyte (TB)—impractical for general use (Table 1). In contrast, TIPars took only 21 seconds on a 64-core server and required only about 1.4 gigabytes (GB) peak memory usage while those of MAPLE were 33 seconds but 24.04 GB. Overall, TIPars ran 12- and 400-fold faster than EPA-ng and IQ-TREE2 respectively, with 98.6% to 99.9% less memory used, an efficiency comparable to that of the leading program, UShER (using 2 seconds with 64 cores). Another computational performance comparison was performed on smaller dataset with 800 bacterial 16S rRNA sequences (16S), for which RAPPAS and PAGAN2 were successfully run with 6 and 33-fold total run times longer than TIPars respectively and is included in S1 Table. MAPLE ran slowest in 16S dataset (S1 Table) which might be expected as MAPLE is designed for highly similar sequences and may break down in divergent scenario [13]. For the scalability of large number of queries, TIPars took 21 hours and 24 minutes with 17.5 GB peak memory usage to insert 200,000 genome sequences into the SARS2-100k reference tree while that of UShER is 9 hours and 35 minutes with 6.7 GB (S2 Table).

Note that neither ancestral sequence reconstruction nor query sequence alignment are included in the runtimes of TIPars in Table 1. To pre-compute the ancestral sequences of SARS2-100k dataset (30,511 internal nodes in the reference tree), it took nearly 4 hours using PastML v1.9.34 with 64 cores (S3 Table). The time required for aligning 100 SARS-CoV-2

genomes to the reference sequence (hCoV-19/Wuhan/WIV04/2019|EPI_ISL_402124) using MAFFT v7.505 (—addtotop) [14] to generate the alignment of query samples (as done in GISAID [1]) for general usage of TIPars (S3 Table) was 0.6 second.

## Single taxon placement

Adding a single sequence sample into a reference tree (single taxon placement) is the basic step for expanding a phylogeny with new sequences. We tested TIPars, UShER, EPA-ng, RAP-PAS and APPLES-2 on four datasets, the SARS-CoV-2 genomes (SARS2-100k), 16S ribosomal RNA genes (16S), hemagglutinin genes of human seasonal influenza A viruses (H3N2) and Newcastle disease virus genomes (NDV). The average pairwise genetic distances (substitutions per site) of SARS2-100k and H3N2 were less than 0.04 ("similar sequences"), while those of 16S and NDV were greater than 0.12 ("divergent sequences") (S4 Table). For the SARS2-100k dataset, EPA-ng, RAPPAS and APPLES-2 were not tested due to infeasibly large memory requirements and/or long runtimes.

Based on the post-order traversal, we selected one sequence between every ten taxa from the SARS2-100k sequence alignment, resulting in 9,602 sequences, i.e., 10% of all taxa in the tree. These selected sequences were individually removed from the reference tree and multiple sequence alignment (MSA) one at a time and used as the query sample for single taxon placement (leave-one-out [15]). In 16S, H3N2 and NDV datasets, 10% taxa were removed individually and used for the placement test, as done in SARS2-100k. Ancestral sequences and mutation-annotated tree [7] were re-constructed to TIPars and UShER for each leave-one-out test respectively, except in SARS2-100k dataset due to the extremely large computational requirement (S3 Table).

To evaluate the accuracy of each single taxon placement, we first calculated the Robinson-Foulds (RF) distance [16] between the reference tree before the taxon removal and the resulting tree after the placement for each program. RF distance is a measure of the topological clustering difference between two trees. A RF distance of zero indicates that the two trees are topologically identical; that is, the single taxon placement position is the same as the original position—a true positive (TP) outcome.

With the aid of ancestral information and MSA of full sequences, TIPars performed accurately on phylogenies derived from highly similar (SARS2-100k and H3N2) and divergent (16S and NDV) sequences, yielding highest proportions of TP in three out of four datasets, except in H3N2 where UShER was the best (Fig 1B). TIPars had smaller or equal mean RF distance than others in NDV and SARS2-100k while EPA-ng and UShER were the lowest in 16S and H3N2 respectively (Fig 1C). A decrease in accuracy on more divergent sequences was observed from UShER, perhaps because UShER was only based on the sequence mutations to a particular reference sequence as input and thereby lost insertion information [7]. The performance of RAPPAS ([65.0%, 71.3%]; Fig 1B) was close to EPA-ng that performed stably on different similarity datasets ([81.3% to 83.8%]; Fig 1B). APPLES-2 ran fast but with a lower accuracy, as appeared to be consistent with other studies [17,18] (Fig 1B and 1C). A test of all taxa removed individually for single taxon placement in 16S, H3N2 and NDV datasets is presented in S1 Fig and the results were consistent with those in Fig 1B–1D. Note that in this another test, the ancestral information has not been reconstructed for TIPars and UShER for each leave-one-out test due to the large computations that would cause a bias for their accuracies (S1 Fig).

In addition, we noted that due to the massive sequencing of SARS-CoV-2 by different research groups, sequencing quality varies and ambiguous bases often occur in consensus genome sequence data, both factors that could affect placement accuracy. To account for

ambiguity in sequencing data, we used a specific substitution scoring table based on the IUPAC nucleotide ambiguity codes (and equivalently for amino acids; S5 and S6 Tables) for the taxa placement and insertion processes (details in Methods), which achieved a robust performance on sequences of different quality.

Notably, when searching through the whole phylogeny for the best position to place a taxon, there may be cases in which multiple branches achieve equal minimum substitution scores (multiple placements). In the case of inserting multiple sequences, multiple placements happened in each insertion of query sequences would result in increasing number of possible trees for expansion. TIPars produced a lower number of multiple placements than other tools in three out of four test datasets, except in H3N2 where UShER had smallest (APPLES-2 was not compared with because it only calculates one placement for one query [11]) (Fig 1D). For the SARS2-100k dataset, TIPars generated 23% fewer multiple placements than UShER (21.3% vs 27.7%; Fig 1D).

To assess the practicability for extremely large phylogenies, we applied TIPars and UShER in single taxon placement test over the global SARS-CoV-2 phylogenetic tree with 659,885 masked genome sequences (SARS2-660k) downloaded from the Global Initiative on Sharing All Influenza Database (GISAID) [1] on September 6, 2021. 65,989 sequences (10% of the total taxa in the tree) were removed individually and re-inserted. The ancestral information has not been reconstructed to both TIPars and UShER for each leave-one-out test. Cumulative proportions of single taxon placement results are shown in Fig 1E with different RF distance cut-offs. TIPars produced trees with significantly higher topological similarity to the reference tree, with a median RF distance of 0.5 and mean of 5.8 (99% confidence interval (CI) = [5.5–6.1]), compared to UShER (median RF distance is 3.0 and mean is 31.2 (99% CI = [30.0–32.4])) at 99% significance level (p-value $< 10^{-10}$).

A potential caveat in the SARS-CoV-2 benchmarks is that ancestral sequences (required by TIPars) and mutation annotated trees (required by UShER), once initially pre-computed with the full dataset, would not be reconstructed in each leave-one-out, because of high computational cost. However, by comparing the benchmarks in 16S, H3N2 and NDV datasets without (S1 Fig) and with (Fig 1B to 1D) ancestral information reconstruction for them, with an observed reduction of accuracy from ≥90% to 81%-94%, TIPars showed better placement accuracy than UShER in two out of three datasets with an exception of H3N2 in both scenarios.

## Multiple taxa insertion

Multiple taxa insertion is an alternative method in determining the phylogenetic position of new sequences over conventional *de novo* phylogeny construction. TIPars and four other programs (IQ-TREE2, PAGAN2, UShER and MAPLE) were applied for a comprehensive evaluation of performance.

In the SARS2-100k dataset, we performed multiple taxa insertion for 100 sets of 100 and 1000 randomly selected sequences (an example is shown in Fig 2A), namely random100 and random1000, and 100 sets of 100 and 1000 successively selected sequences (i.e., a set of taxa selected in turn following the tree post-order traversal (leave-many-out); an example is shown in Fig 2B), namely successive100 and successive1000. From each of the 16S, H3N2 and NDV datasets, 100 sets of 50 sequences were randomly selected. The selected sequences were pruned from the corresponding reference tree and designated as multiple taxa to be reinserted for the corresponding tests. Ancestral sequences (used in TIPars) and mutation-annotated tree (used in UShER) were re-constructed for each leave-many-out test respectively, except for SARS2-100k dataset.

RF distance and tree log-likelihood were used to evaluate the performance of the multiple taxa insertion. To evaluate the topology accuracy, the tree produced by the four programs were compared to the original reference tree (leaf taxa unpruned) to obtain the RF distance.

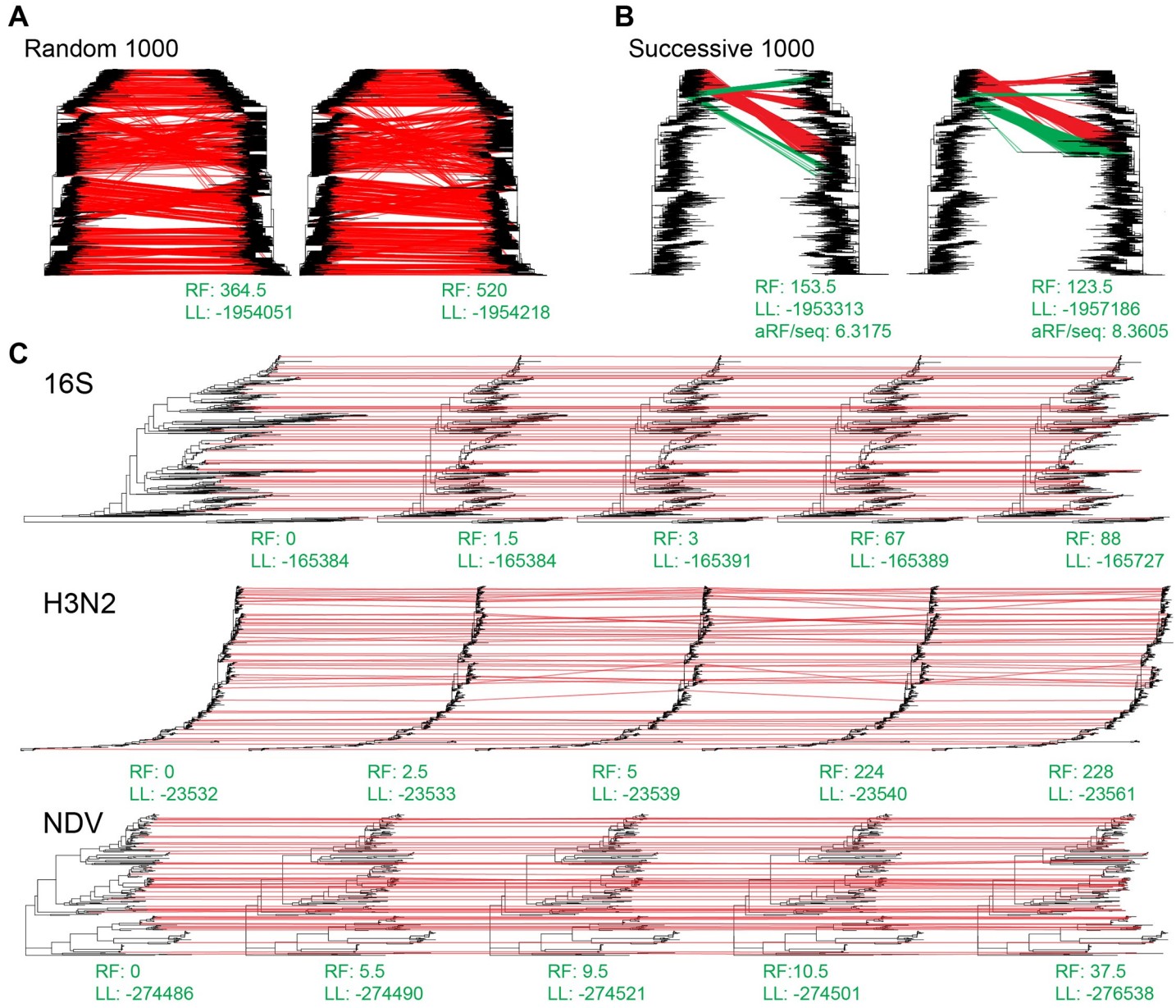

**Fig 2. Taxa insertion visualization.** (A) Example of TIPars (left) and UShER (right) resulting trees paired with the reference SARS2-100k reference tree (left tree in each tree comparison) for the insertion of 1000 randomly selected taxa sequences. Red lines link the corresponding positions of inserted taxa between reference and resulting tree. Green lines indicate different taxa insertion positions between TIPars and UShER. Averaged RF distance per sequence (aRF/seq) compared to the reference tree is shown at the bottom. (B) Example of TIPars (left) and UShER (right) resulting trees paired with the reference SARS2-100k reference tree (left tree in each tree comparison) for the insertion of 1000 successively selected taxa sequences. (C) Demonstrations of the resulting trees for 50 randomly selected taxa in NDV, 16S (midpoint rooted) and H3N2 datasets. From left to right are trees of reference, TIPars, UShER, IQ-TREE2 and PAGAN2. RF distances (RF) compared to the reference tree and Gamma20 log-likelihoods (LL) are shown below each tree.

Gamma20 log-likelihoods (LL) of the resulting trees after optimizing the branch length were also computed using FastTree2 v2.1.11 (double-precision version) and their differences were used for evaluation.

For the random100 and random1000 datasets, only analyses using TIPars and UShER were tested. The resulting trees from multiple taxa insertion using TIPars had significantly smaller

RF distances than those generated using UShER (negative values in Fig 3A). In addition, LLs from TIPars were significantly higher than those of UShER (positive values in Fig 3B). Moreover, TIPars resulting trees tended to be very close to the reference tree as indicated by smaller LL differences (S2A and S2B Fig). A demonstration of the taxa-insertion of 1000 samples is

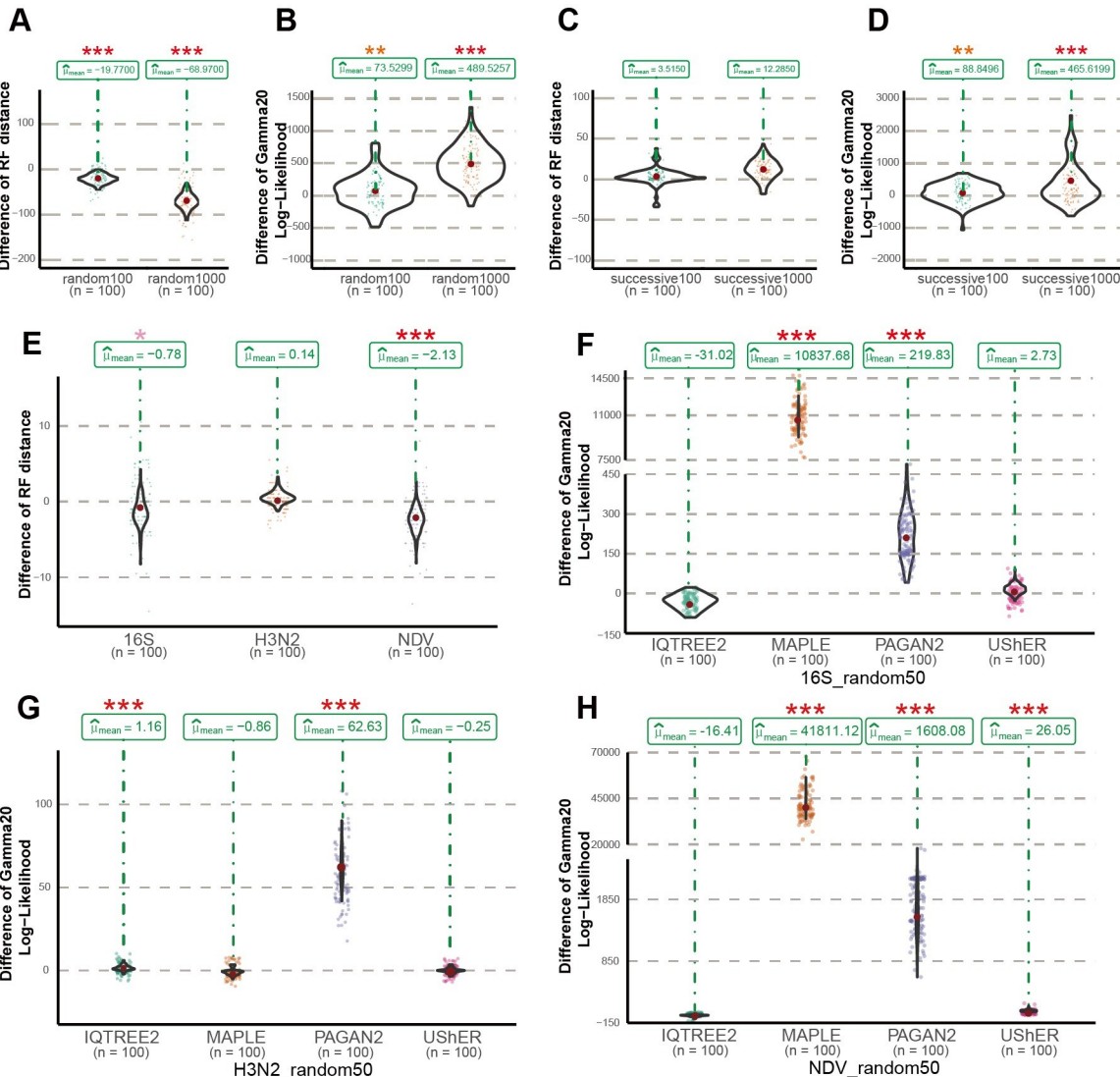

**Fig 3. Multiple sequences insertion performance.** (A) Violin graphs showing the distribution of paired differences of the RF distances between the resulting trees generated by TIPars and UShER (TIPars – UShER) for the random 100 and 1000 multiple sequences insertions. (B) Violin graphs showing the distribution of paired differences of the Gamma20 log-likelihoods between the resulting trees generated by TIPars and UShER (TIPars – UShER) for the random 100 and 1000 multiple sequences insertions. (C) Violin graphs showing the distribution of paired differences of the RF distances between the resulting trees generated by TIPars and UShER (TIPars – UShER) for the successive 100 and 1000 multiple sequences insertions. (D) Violin graphs showing the distribution of paired differences of the Gamma20 log-likelihoods between the resulting trees generated by TIPars and UShER (TIPars – UShER) for the successive 100 and 1000 multiple sequences insertions. (E) Distribution of paired differences in the RF distances between the resulting trees generated by TIPars and UShER (TIPars – UShER) on 16S, H3N2 and NDV random 50 multiple sequences insertions. (F) Distribution of the paired differences in the Gamma20 log-likelihoods between the resulting trees generated by TIPars and the four other programs (TIPars – Others) on 16S random 50 multiple sequences insertions. (G) Distribution of the paired differences in the Gamma20 log-likelihoods between the resulting trees generated by TIPars and the four other programs (TIPars – Others) on H3N2 random 50 multiple sequences insertions. (H) Distribution of the paired differences in the Gamma20 log-likelihoods between the resulting trees generated by TIPars and the four other programs (TIPars – Others) on NDV random 50 multiple sequences insertions. P-values for the right-sided paired t-tests are indicated by the asterisk on top of each violin diagram, where $p<0.05$ is indicated by one pink asterisk (*), $p<0.01$ by two orange asterisks (**) and $p<0.001$ by three red asterisks (***).

illustrated in Fig 2A. We observed more crossing lines from reference tree to the UShER resulting tree, suggesting more misplaced insertions.

As for 16S, H3N2 and NDV datasets, TIPars achieved significantly smaller RF distances than UShER in 16S and NDV while no significant difference was in H3N2 (Fig 3E). RF distance was not available for IQ-TREE2 and PAGAN2 because they may alter the input tree topology after inserting sequences causing large RF distance of inserted trees against the reference tree (S7 Table) [2]. For the tree log-likelihood, TIPars outperformed UShER in NDV with significantly higher LLs of resulting trees while there was no significant difference between them in 16S and H3N2 datasets (Fig 3F to 3H). IQ-TREE2 had the highest mean LLs in 16S and NDV datasets and PAGAN2 was the lowest in all datasets (Figs 3F to 3H and S3). MAPLE outperformed other methods in H3N2 dataset with highest Gamma20 log-likelihoods (LL) of the resulting trees (Fig 3G) while a significant drop of LLs was observed in more divergent datasets 16S and NDV (Fig 3F and 3H). The RF distances from MAPLE resulting trees in 16S, H3N2 and NDV datasets are comparatively large and presented in S7 Table. This may be explained by the fact that MALE is designed for highly similar sequences and may break down in divergent scenario [13]. MAPLE implemented a local phylogenetic placement algorithm where the 'best' placement of a query sequence is searched on a subset of branches in the tree while TIPars and UShER are global search by a transversal of all branches. Phylogenetic tree is not a well classified data structure (such as red–black tree), a local search may stop too early to retrieve the global best placement especially for divergent phylogeny which contains long branches. A visualization of taxa insertion is presented in Fig 2C where the resulting tree of TIPars was the closest to the reference tree with the smallest RF distance.

For the successive100 and successive1000 datasets, TIPars resulting trees had significantly larger RF distances than those of UShER (Fig 3C). However, the log-likelihoods of the TIPars resulting trees were significantly higher than those of UShER (Figs 3D, S2C and S2D). By comparing the trees generated from TIPars and UShER (Fig 2B), we discerned that TIPars inserted some query taxa (green lines in Fig 2B; successive taxa pruned from the reference tree) into two subtrees, wherein one (containing over half these queries) had the same topology as the reference tree, while UShER inserted these queries mostly within a monophyletic clade different from the reference tree. As a result, UShER retained the local topology (better RF distance) [19,20] but missed the global topology (worse LL). By performing RF distance comparison for each query taxon, rather than on aggregated data (detailed in Methods), we found that the RF distances (per query taxon) resulted from UShER were not significantly smaller than those of TIPars (S8 Table). While not a reliable metric for these 'successive' datasets, RF distance may be suitable for comparing the performance of taxa insertions in scenarios, such as the random100 and random1000 tests, that are similar to single taxon placement cases in which most removed taxa are within different monophyletic clades due to randomness [21].

Given the fact that the absolute number of mutations are reported as branch lengths of the UShER's output tree [7] (details in Methods), to make the log-likelihoods of the resulting trees of UShER and TIPars comparable, we applied FastTree2 to reoptimize the branch lengths with fixed topology [3]. As a result, 73% of resulting trees from TIPars had higher LL values than UShER (S9 Table). Notably, compared to taxa insertion (Table 1), the re-optimization was far more time-consuming. For example, optimization for a SARS2-100k tree took 10 to 12 hours and required around 126 GB peak memory (S10 Table).

## Tree refinement

For multiple sequence insertion, a phylogenetic placement based method, such as TIPars and UShER, does not alter the backbone tree topology. We further tested the effect of tree

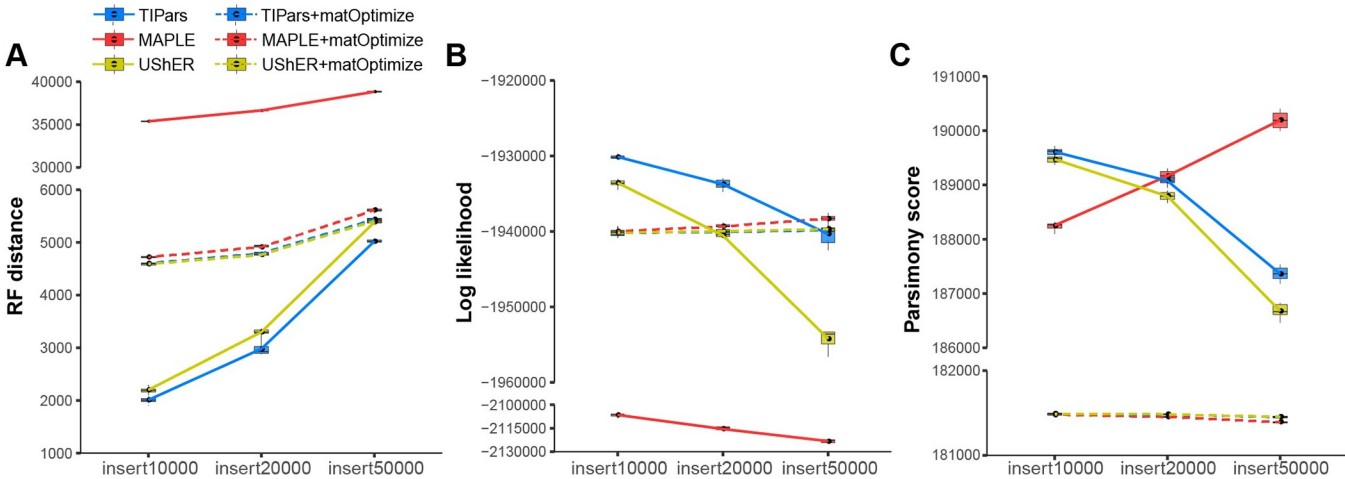

**Fig 4. Multiple sequences insertion with tree topology refinement.** (A) RF distances between inferred trees and the reference tree for 10 leave-out tests of 10000, 20000 and 50000 taxa in SARS2-100k dataset respectively. Smaller values correspond to more similar to the reference tree. 'X+matOptimize' indicates the refined trees of X with tree topology optimized by matOptimize. (B) Log-likelihoods (LK) (computed by FastTree2 (double-precision version)) of phylogenies inferred by different methods. Larger values represent more likely estimates. (C) Parsimony scores (computed by UShER) of phylogenies inferred by different methods. Smaller values represent smaller number of substitutions needed for each site.

refinement on the inserted trees from TIPars, UShER and MAPLE by matOptimize [22] which applies fast subtree pruning and regrafting (SPR) moves to improve tree topology. We compared tree log-likelihood (LK) calculated by FastTree2 (double-precision version) (Fig 4B) instead of Gamma20 Log-likelihood (LL) because it returns '-nan' for LLs of inserted trees from MAPLE. But we present the LL results for TIPars, UShER and those with topology refinement by matOptimize in S4 Fig.

For 10 leave-out tests of 10000 (insert10000), 20000 (insert20000) and 50000 taxa (insert50000) in SARS2-100k dataset without tree refinement, TIPars achieved smallest RF distance and highest LK of inserted trees in all tests (Fig 4A and 4B). UShER had the smallest tree parsimony score in insert20000 and insert50000 tests while MAPLE had the smallest parsimony score in insert10000 but an increased parsimony score was observed when inserting more queries (Fig 4C).

After topology refinement, the refined trees had significant smaller parsimony scores but larger RF distance for trees of TIPars and UShER (Fig 4A and 4C). The inserted trees from TIPars had decreased LKs after tree refinement in insert10000 and insert20000 and similar LK values were observed in insert50000 before and after tree refinement (Fig 4B). A possible reason is that matOptimize is fully parsimony-based method without a promise on the tree log-likelihood [22]. Accounting for the larger RF distance, this may because our provided SARS2-100k reference tree was built by a maximum likelihood method (IQTREE2), so the RF distance after refinement by matOptimize may not become smaller. The refined trees from MAPLE had smaller RF distances, larger LKs and smaller parsimony scores compared to its inserted trees which suggests a tree refinement can be useful for the inserted tree of MAPLE.

## Inserting novel sequences

To verify the practicability of TIPars to add novel sequences into a given phylogeny, we insert additional SARS-CoV-2 sequences into the SARS2-100k reference tree which were not included in the SARS2-100k dataset. We randomly selected SARS-CoV-2 sequences from GISAID that were collected before January 1, 2021, the same timepoint as sequences for generating SARS2-100k reference tree. Twenty sets of 100, 1000, 5000 and 10000 genome sequences were generated as the queries for taxa insertion using TIPars and UShER.

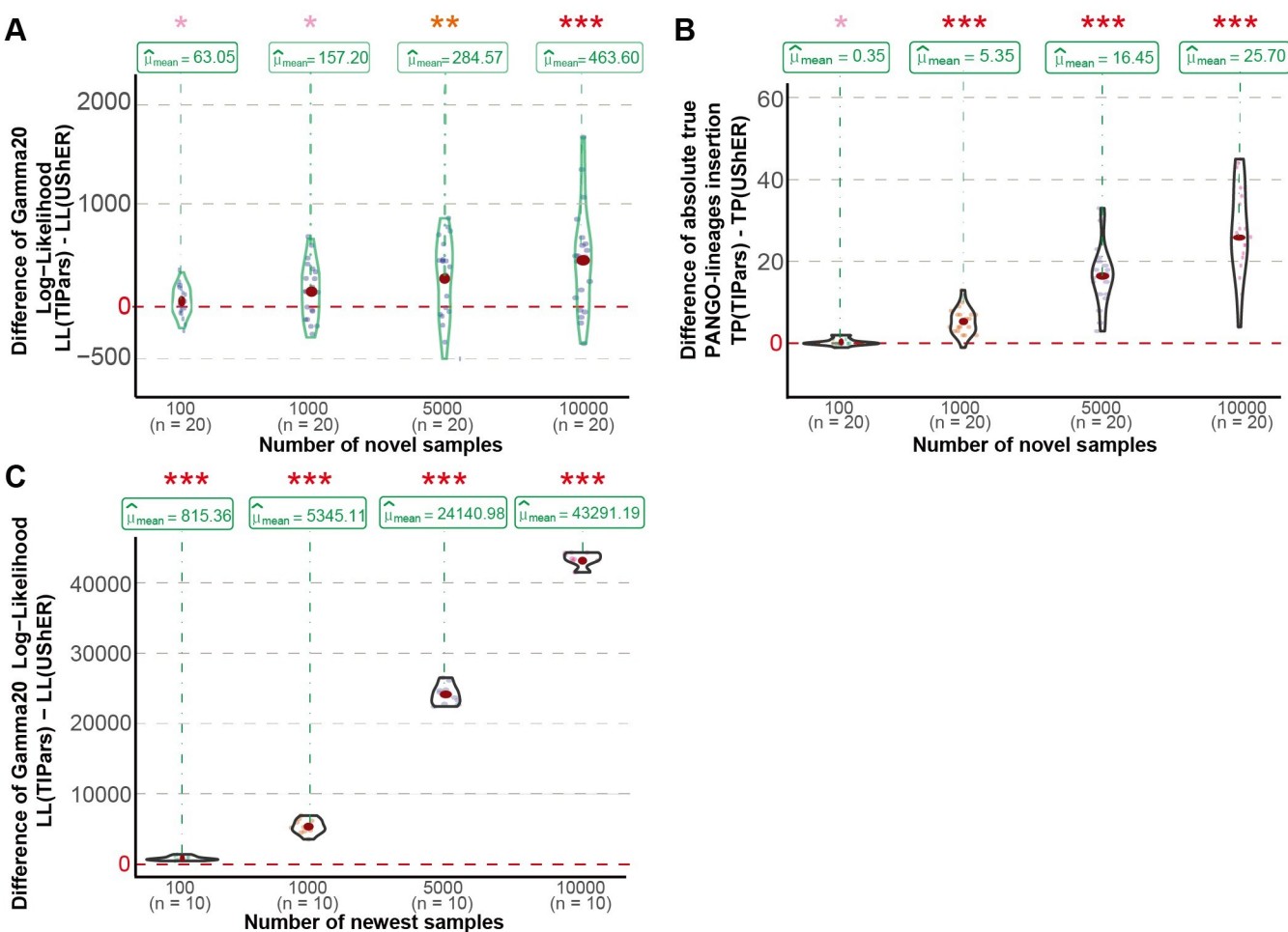

**Fig 5. Performance of inserting novel real-world sequences.** (A) Violin graphs present the distributions of the paired differences between TIPars and UShER with respect to Gamma20 log-likelihood (LL) for insertion of novel sequences collected before January 1, 2021, the same timepoint as sequences for generating SARS2-100k reference. (B) Violin graphs present the distributions of the paired differences between TIPars and UShER with respect to the absolute number of true PANGO-lineages insertion (TP) for insertion of novel sequences collected before January 1, 2021. (C) Violin graphs present the distributions of the paired differences between TIPars and UShER with respect to LL for insertion of newest sequences submitted to GISAID in between January 1 2022 to June 4 2022. P-values for the right-sided paired t-tests are indicated by asterisks above each violin plot, where p<0.05 is indicated by one pink asterisk (*), p<0.01 by two orange asterisks (**) and p<0.001 by three red asterisks (***).

Log-likelihoods of the resulting trees from each program were calculated and their pairwise differences between TIPars and UShER were used to evaluate the performance. RF distance was not a suitable metric in this experiment because a comparable reference tree was not available. TIPars provided resulting trees with significantly better Gamma20 log-likelihoods than UShER in all situations (p-values <0.05; Fig 5A).

We also compared the PANGO lineages (PANGOlins) assignment of the added samples [23] to validate the accuracy. Only PANGOlins that existed in the reference tree were considered. We assigned each newly inserted sequence with the lineage name of the subtree under the parental node of the inserted position. The subtree was annotated by its descendant reference taxa if all of them were monophyletic [24]. A true positive was defined as when the assigned lineage of a query sequence was identical to its original PANGOlins label. We ignored queries within unannotated subtrees in the calculation. TIPars outperformed UShER by achieving significantly higher true positive samples on all datasets with an average of 92%

PANGOlins accuracy under a right-tailed paired t-test (p-values < 0.05) (Fig 5B and S11 Table).

SARS-CoV-2 sequences submitted to GISAID between January 1, 2022 and June 4, 2022 were used to verify TIPars's performance for genome sequences potentially featuring new yet unseen mutations from the reference sequence data (Method). Fig 5C demonstrates that TIPars achieved significantly higher Gamma20 Log-likelihoods than UShER in 10 sets of 100, 1000, 5000 and 10000 sequences inserting to the SARS2-100k reference tree. Their comparison to the "true tree" reconstructed by FastTree2 v2.1.11 (double-precision version) could be checked in S4 Fig. This suggests that although the performance of TIPars was better than UShER, more thorough statistical phylogenetic optimization is recommended if the target is a high likelihood tree.

## Discussion

Maximum likelihood (ML) and Bayesian statistical inferences using probabilistic models are the mainstream methods for ancestral sequence estimation [9,10,25,26]. Several studies have shown that the ML generated ancestral sequences may reveal more accurate (especially intermediate) evolutionary information than the 'mutation-annotated tree' inferred by Fitch–Sankoff parsimony algorithm [27], as implemented in UShER [26,28,29]. The evolutionary information can be used to distinguish insertion, deletion and substitution events in the searching of taxon placement [30], which may explain the robustness of TIPars on more divergent phylogenies [8]. Overall, compared to existing phylogenetic placement programs, TIPars showed better placement of taxon and accuracy of insertion in the different phylogenies with homogenous (H3N2 and SARS2-100k) and divergent (16S and NDV) sequences, and even extremely large phylogeny (SARS2-660k) (Figs 1 and 3). A potential caveat to be considered is that ancestral sequences (TIPars) or mutation-annotated tree (UShER) once initially pre-computed with the full dataset would not be re-computed in each leave-one-out and leave-many-out tests for SARS-CoV-2 datasets because of high computational cost. The performance of both TIPars and UShER could have been benefited by the information of the query sequence remanent in these initially pre-computed ancestral sequences or mutations. However, the benchmark of inserting novel sequences on SARS2-100k dataset (Fig 4), which carry no information from the queries, could verify a superiority of TIPars over UShER. Meanwhile, TIPars achieved reasonable runtime and memory usage (Table 1). Although UShER had lower accuracy in the divergent sequence datasets (16S and NDV; Figs 1 and 3), it ran faster than TIPars for sequence insertion step (Table 1).

For EPA-ng performance, we considered the potential noise for the single taxon placement (Fig 1B and 1C) due to the conversion from the original polytomy to binary by generating zero branches for EPA-ng input using 'multi2di' function in Ape package and the conversion from the resulting binary tree to polytomy for RF distance computation using 'di2multi' function (default tolerance = 1E-8). Note that the pendant length P-Q would not change the bipartition of the tree, and thus would not influence the RF distance. We found there are 82, 304 and 2 cases with both distal and sibling branch lengths less than 1E-8 (i.e., EPA-ng inserts the query into a zero branch (A-B) generated by the first 'multi2di' processing) for 16S, H3N2 and NDV datasets respectively, which have been re-collapsed by 'di2multi' (using the default tolerance of 1E-8) (S12 Table). The high true positive rates of them (79/82≈96% in 16S, 299/304≈98%, 2/2≈100%) implies limited negative influence from the two conversions (first 'multi2di' and then 'di2multi') on EPA-ng's performance.

Considering both placement accuracy and RF distance (Fig 1B to 1C), we would summarize that EPA-ng, UShER and TIPars performed best in 16S, H3N2 and NDV datasets respectively

(S13 Table). This was consistent with the theoretical expectations to a certain extent, as EPA-ng could be considered as a fully likelihood-based method, UShER as a fully parsimony-based and TIPars as a combination of likelihood (in ancestral reconstruction) and parsimony (in scoring candidate branches for placement/insertion) approaches. It is reported that the larger the sequence divergence is, the larger the inconsistency on the topology between the trees built using between likelihood- and parsimony-based methods [31]. A consistent observation was in multiple taxa insertion test when considering both tree log-likelihood and RF distance for the performances of TIPars, UShER and IQTREE2 as presented in Fig 3E to 3H where IQTREE2 is a fully likelihood-based method (S13 Table). TIPars ranked second in all datasets while IQTREE2 ranked the best with the highest mean tree log-likelihoods in both 16S and NDV datasets. A caveat is that we have not compared IQTREE2 with RF distance because it may alter the input tree topology after inserting sequences causing large RF distance of inserted trees against the reference tree (not exactly based on phylogenetic placement) (S7 Table), which would lead to a bias to its high tree log-likelihood.

Although we showed that TIPars resulting trees had higher tree log-likelihoods compared to other programs in most tests, a general limitation of the phylogenetic placement method is that errors from incorrect placements accumulate as multiple sequences are inserted sequentially. In order to minimize the errors due to large numbers of sequence insertions, we suggest conducting tree refinements on not only branch length but also tree topology using different techniques such as nearest-neighbor interchanges (NNIs) and subtree-pruning-regrafting (SPRs) [3]. Fig 4 shows the accuracy performance of TIPars when expanding the tree to 111% (insert10000), 125% (insert20000) and 200% (insert50000) with and without tree topology refinement. Users could determine the frequency to do a tree refinement according to their needs. After a topology refinement, the ancestral sequences should be reconstructed again for TIPars which will take additional computation cost.

We appreciate the contribution of UShER tool suite (including the latest update to UShER-sampled) for maintaining the huge SARS-CoV-2 phylogenetic tree with millions of genome sequences in near real time [32]. UShER is the fastest using the smallest memory among our tested tools (Table 1) and the absolute difference would become larger when dealing more query samples (S2 Table). In contrast, TIPars performed more robust in phylogenies with both similar and divergent sequences than UShER (Figs 1 and 3) with feasible runtime and memory usage (S2 Table).

## Materials and methods

### Implementation of TIPars

After assigning the ancestral sequences at every internal node and taxa sequences at external nodes, TIPars inserts a set of new samples into the reference phylogenetic tree sequentially based on parsimony criteria.

For a query sequence Q, TIPars computes the minimal substitution score against all branches in the tree. While inserting query Q into a branch A-B (parent node—child node) at a potential newly added node P (Fig 1A), the substitution score is the sum of mutations by which query Q differs from both node A and node B based on a specific substitution scoring table (based on the IUPAC nucleotide ambiguity codes for nucleotides, S5 Table or the BLOSUM62 scoring matrix [33] for amino acids, S6 Table). For example, considering the mutations for inserting Q (ACG**T**) into the branch between node A (ACC**G**) and node B (ACG**C**) (Fig 1A), there is only 1 mutation that is at the 4th site of Q where the genetic character '**T**' differs to both characters 'G' in node A and 'C' in node B. The single branch with the minimum substitution score $\sigma$ is reported as the best placement.

However, in terms of multiple placements where more than one branch has the same minimum substitution score, TIPars applies simple but practical rules to filter them to a single best placement such that multiple queries would be inserted sequentially based on one resulting tree. The first priority is to select the branch with node A containing the most numbers of child nodes that is to place the query at a larger subtree. The second priority is to select the branch with node A of the lowest node height (i.e., the total branch length on the longest path from the node to a leaf [34]) that is to place the query at an ancestor with less accumulated mutations. Finally, in the case where the ambiguity cannot be resolved by the first two priorities, TIPars randomly selects one. Even though TIPars will filter out multiple placements, these potential placements will also be printed to notify the user.

We proposed a local estimation model to calculate the pendant length of the newly introduced branch P-Q ($l_{P-Q}$) which considers the branch lengths of the local triplet subtree (A,(B, Q)) (Fig 1A). Pendant length is defined as $l_{P-Q} = \sigma/(\delta_A+\delta_B)^*l_{A-B}$, where $\delta_A$ and $\delta_B$ are the unique mismatch substitution scores of Q to A and B, respectively, $l_{A-B}$ is the original length of branch A-B and $\sigma$ is the minimal substitution score. The location of P on branch A-B is determined by the ratios of $\delta_A$ and $\delta_B$, i.e., distal length: $l_{A-P} = \delta_A/(\delta_A+\delta_B)^*l_{A-B}$, and sibling length: $l_{B-P} = \delta_B/(\delta_A+\delta_B)^*l_{A-B}$. The ancestral sequence of node P is estimated by majority vote of the nucleotide bases of sequence A, B and Q. To retain the reference tree topology, a potential nucleotide base of P is derived only from A or B. For the special case where $l_{A-B}$ is zero but $\sigma$ is not, TIPars considers the upper branch of A's parent to A for scaling.

We implemented TIPars using Java with the library BEAST [34]. Both FASTA and VCF formats are acceptable for loading sequences, and NEWICK format is for the tree file. FASTA file is the default setting, but VCF file is more memory-efficient for large datasets of high similar sequences, e.g. SARS-CoV-2 viral genomes. To convert a FASTA file to VCF format with all sequence mutations, i.e. insertion, deletion and mismatch, we used a Python package PoMo/FastaToVCF.py [35].

### Preparation of benchmark datasets

Unmasked SARS-CoV-2 MSA and metadata from GISAID was downloaded on July 6, 2021. All SARS-CoV-2 viral genome sequences collected before January 1, 2021 were extracted from the MSA. To ensure the sequences used for downstream analysis were complete, we removed SARS-CoV-2 genomes with sequence length < 29,000 bp and > 0.5% Ns (namely GISAID202101). To ensure that the global phylogenetic diversity was well represented in the subsampled dataset, sequences from all lineages as designated by the PANGO nomenclature system [23] were subsampled where PANGO lineage labels were extracted from the metadata. Where fewer than 50 sequences of a given lineage were found in the global dataset, all sequences of the lineage were included. This resulted in a final sub-sampled dataset of 96,020 sequences from 1,249 PANGO lineages, with hCoV-19/Wuhan/WIV04/2019/EPI_ISL_402124 included as the reference genome (namely SARS2-100k). The SARS2-100k reference tree was then built using IQ-TREE2 (GTR model) using the EPI_ISL_402124 as the root which took around 14 hours using 40 Intel Xeon Gold 6242 CPU cores. Ancestral sequences of each internal node were estimated using PastML v1.9.34 (maximum parsimony method with ACCTRAN model) with the MSA and the IQ-TREE2-generated tree as input (S3 Table).

ACCTRAN model estimates the ancestral state changes to be as close to the root as possible which provides a non-random choice when multiple best parsimonious solutions exist and might introduce bias in prioritizing reverse mutations. Other models such as DOWNPASS and DELTRAN are available in PastML that can be chosen as alternative.

Three small representative nucleotide sequence datasets, namely, bacterial 16S rRNA genes (16S), hemagglutinin genes of human seasonal influenza A viruses (H3N2), and Newcastle disease virus genomes (NDV), were prepared for the comparison of programs performance. The 16S dataset was downloaded from Genomes OnLine Database [36] and randomly down-sampled to 800 sequences. HA sequences of 800 H3N2 viruses were randomly extracted from GISAID [1]. The 235 NDV sequences were downloaded from GenBank. Alignments were constructed using MUSCLE [37]. Reference trees of these datasets were built using RAxML [38] standard hill-climbing heuristic search with 100 multiple inferences and GTRGAMMA model. Ancestral sequences were estimated using HyPhy 2.5 (ML joint method) [25,26]. The runtimes and memory usages of these processing are shown in S14 Table.

## Novel SARS-CoV-2 query sequence dataset

To generate novel query sequences for the 20 sets of 100, 1000, 5000 and 10000 sequences, SARS-CoV-2 genomes that were not included in the SARS2-100k dataset were randomly selected from the GISAID202101 dataset. Selected sequences were then aligned to the SARS2-100k sequences alignment by opening necessary gaps to obtain the full-length MSA. The newly selected sequences were extracted to obtain the final query sample sets. Corresponding new gaps were also added back to the ancestral sequence alignment for each dataset generated. PANGO lineages data for the novel SARS-CoV-2 query sequences and taxa of the reference tree were downloaded from GISAID on July 6, 2021.

3,947,326 sequences submitted between January 1 and June 4, 2022 were taken from the full multiple sequence alignment downloaded from GISAID on 6th June 2022. 992 PANGO lineages were found in the dataset, with 87% of sequences from the BA lineages (the Omicron variant) (S15 Table). To reduce bias from the overwhelming variants, a maximum of the newest 20 sequences (in terms of submission date) were taken from any PANGO lineage. PANGO lineages were randomly selected and the newest sequences from them were added to the sample pool to generate datasets of 100, 1000, 5000 or 10000 sequences for insertion into the tree. New PANGO lineages were selected until the requisite sample size was obtained with always the newest sequences taken first. This process was repeated 10 times.

## Benchmark programs

We compared TIPars to 7 state-of-the-art phylogenetic placement tools: UShER, EPA-ng, APPLES-2, IQ-TREE2, RAPPAS, PAGAN2 and MAPLE. EPA-ng, APPLES-2 and RAPPAS only place a single taxon, and IQ-TREE2, PAGAN2 and MAPLE were only used to insert multiple taxa.

For the SARS2-100k dataset, only TIPars, UShER and MAPLE were considered, as the other programs were not able to complete the computation within a reasonable runtime and memory usage (Table 1). For the three smaller datasets, we compared all programs. To make it fair, we ran MAPLE without topology refinement since TIPars and UShER will not alter the backbone tree for multiple taxa insertion. Details of the commands used for different programs can be found in S16 Table.

TIPars, UShER, EPA-ng and RAPPAS reported multiple placements for single taxon insertion. The default marked "best" placements by TIPars and UShER of each individual taxon were used for our accuracy evaluation. EPA-ng and RAPPAS reported their results sorted by log-likelihood, so the placement with the highest log-likelihood was used in the assessment.

As EPA-ng, RAPPAS and PAGAN2 only accept binary trees, the original polytomous tree was converted to a binary tree for input and the resulting trees after performing the insertion were re-collapsed to polytomy using the Ape R package [39] before evaluation of their

accuracy (RF distance against the original reference polytomous tree). Results from TIPars, UShER, IQ-TREE2, APPLES-2 and MAPLE are based on the same benchmark reference polytomous trees. RAPPAS was run by integrating with RAxML-ng v1.1.0 [40] as in [12,41].

### Evaluation metrics

For single taxon placement evaluation, we first pruned one taxon from the reference tree and then re-inserted it. To assess consistency between placement algorithms and the typical tree construction approach, we proposed using Robinson–Foulds (RF) distance as a measure of the tree topology accuracy, calculated by TreeCmp [42]. When the RF distance between a resulting tree and the reference tree is zero, the topology of the resulting tree is the same as the reference tree, meaning that the algorithm correctly inserted the query sample into the reference tree.

For multiple taxa insertion evaluation, we randomly pruned a set of taxa from the reference tree as queries and re-inserted them. In addition to using RF distance to compare the resulting tree after adding all query taxa against the reference tree, we calculated the log-likelihood of the resulting tree as a measurement of the accuracy of the taxa insertions. Note that UShER outputs the final constructed tree using the number of mutations as branch lengths, otherwise no branch length would be specified at modified branches [7]. To make the log-likelihood comparable, we applied FastTree2 (double-precision version) to optimize the branch lengths and then compute Gamma20 log-likelihoods for resulting trees and reference tree. Since FastTree2 only accepts binary trees polytomies were converted to binary branches by the 'multi2di' function in Ape R package using default tolerance of 1E-8. On the other hand, another RF distance comparison specific to each query taxon, instead of on aggregate, is defined as follows. First, all queries were reinserted. Next, for every query taxon, we pruned all other queries from the resulting and reference trees (i.e. single taxon placement), and then computed their RF distance. The RF distance per query taxon for statistics is the mean of them in each testing set.

EPA-ng and RAPPAS output the placement information (placed branch, distal length, and pendant length) for a query without construction of the final tree. To compute the RF distance, we converted its output placement information (jplace format [6]) to a resulting tree (newick format) by inserting the query into the reference tree (Fig 1A).

IQ-TREE2 (-g model) and PAGAN2 take the initial tree for a constrained tree search, but they may alter the input tree topology after inserting sequences [2]. The RF distance to the original reference tree is not a suitable metric for these methods; hence it is not given for the single taxon placement test and the RF comparison in multiple taxa insertion tests.

### Statistics

99% t-test confidence intervals and 99% paired t-test p-value (right tail) of the results from TIPars against other programs were computed by Matlab R2013b. All violin graphs were generated in R 4.1.1 using the package ggstatsplot [43]. Illustration and annotation of phylogenetic trees were made using the R package ggtree [44].

### Supporting information

**S1 Fig. Performance for (re)placement of a single taxon.** All taxa were removed individually and used for the placement test for TIPars, UShER, EPA-ng and APPLES-2 in 16S, H3N2 and NDV datasets. Note that the ancestral sequences of TIPars and mutation-annotated tree of UShER have not been reconstructed for each leave-one-out test due to the high computational requirement that would cause a bias for their accuracies. RAPPAS was excluded because of its large computation for the 'pkDB' database. (A) Bars represent the placement accuracy on 16S, H3N2 and NDV datasets. The highest accuracy in each dataset is highlighted in red. (B) Bar

charts representing the mean RF distance calculated from the single taxon placement results on 16S, H3N2 and NDV datasets. The lowest mean RF distance in each dataset is highlighted in red. Panel A and B share the same figure legend in B. (C) Stacked bar charts showing the proportions of single and multiple placement results on 16S, H3N2 and NDV datasets. Proportions with > 0.1% are indicated within the bars.
(TIF)

**S2 Fig. Paired differences of the Gamma20 log-likelihoods (LL) between the result trees and the reference tree (RT) on the SARS2-100k dataset.** (A) Violin plots show the distribution of paired differences for the 100 sets of random 100 multiple sequence insertions. (B) Violin plots show the distribution of paired differences for the 100 sets of random 1000 multiple sequence insertions. (C) Violin plots show the distribution of paired differences for the 100 sets of successive 100 multiple sequence insertions. (D) Violin plots show the distribution of paired differences for the 100 sets of successive 1000 multiple sequence insertions.
(TIF)

**S3 Fig. Paired differences of the Gamma20 log-likelihoods (LL) between the result trees and the reference tree (RT).** (A) Violin graphs show the distribution of the paired differences for 100 sets of 50 random multiple sequence insertions in 16S dataset. The mean difference of MAPLE was 10837.68. (B) Violin graphs show the distribution of the paired differences for 100 sets of 50 random multiple sequence insertions in the H3N2 dataset. (C) Violin graphs show the distribution of the paired differences for 100 sets of 50 random multiple sequence insertions in the NDV dataset. The mean difference of MAPLE was 41811.12.
(TIF)

**S4 Fig. Gamma20 Log-likelihoods (LL) (computed by FastTree2 (double-precision version)) of phylogenies inferred by different methods.** Larger values represent more likely estimates. 'X +matOptimize' indicates the refined trees of X with tree topology optimized by matOptimize.
(TIF)

**S5 Fig. Comparison of Gamma20 log-likelihoods (LL) for the reference tree, and the result trees of TIPars and UShER after inserting newest sequences submitted to GISAID in between January 1 2022 to June 4 2022.** The reference tree was constructed by FastTree v2.1.11 (double-precision version) using the alignment of the newest added sequences and the sequences in SARS2-100k dataset.
(TIF)

**S1 Table. Average run time and memory usage for inserting/placing 50 samples into the 16S 800-taxa reference tree.** Tests were run on a server with 64 Intel Xeon Gold 6242 CPU cores and 1500 GB RAM for 10 repeated runs. The performance of TIPars and UShER were also compared on a general computer with 8 Intel Xeon Gold 6242 CPU cores. TIPars ran with JAVA setting -Xmx1G loaded with FASTA files. "Insertion time" indicates the time for the key step of searching for the best position to place or insert only a query sequence, while "total runtime" is the total time to run the program. EPA-ng insertion time data is not presented as its reporting of this time is limited to whole seconds and given as 0 in this case. MAPLE is a single-thread program indicating by 1 CPU core assigned. N/A indicates that data are not applicable. Total runtimes for the different programs consist of 1) TIPars: "insertion time" (IT) + I/O processing (I/O); 2) UShER: IT + I/O + 'mutation-annotated tree' computation; 3) EPA-ng: IT + I/O + likelihood weight ratio (LWR) computation; 4) RAPPAS: IT + I/O, 5) APPLES-2, IQTREE2 and PAGAN2: their total run times and 6) MAPLE: IT+I/O+'substitution matrix' (without topology refinement). * The run time for RAPPAS to compute its

'pkDB' database (3095.31 seconds on average) is not included in "Mean total run time".
(XLSX)

**S2 Table. Runtime and memory usage for multiple taxa insertion to the 100k-taxa reference tree.** Tests were run on a server with 32 Intel Xeon Gold 6242 CPU cores for 10 repeated runs. TIPars took less than one day to insert 200k SARS-CoV-2 genome sequences into the 100k-taxa reference tree.
(XLSX)

**S3 Table. Average run time and memory usage to align 100 query samples and to reconstruct ancestral sequence for the SARS2-100k dataset.** We estimated the run time and peak memory usage for the two preprocessing steps by TIPars. The first step of query sample alignment is necessary for all other alignment-based phylogenetic placement methods, e.g., UShER, EPA-ng, IQTREE-2 and APPLES-2. For the second step, once ancestral sequences have been reconstructed, they can be reused. Here we performed 10 runs of these two preprocessing steps: 1) aligning 100 SARS-CoV-2 samples to the reference hCoV-19/Wuhan/WIV04/2019| EPI_ISL_402124 using MAFFT (—addtotop); 2) inferring the ancestral sequences for the SARS2-100k dataset using PastML (ACCTRAN method) and present average times. Tests were run on a server with 64 or 8 Intel Xeon Gold 6242 CPU cores.
(XLSX)

**S4 Table. Average pairwise genetic distance per site for each benchmark dataset.** For the genetic distance calculations, random samples of 10% of the SARS-CoV-2 datasets were selected, while all sequences were included for the other datasets.
(XLSX)

**S5 Table. IUPAC code substitution table for nucleotide sequence insertion by TIPars.**
(XLSX)

**S6 Table. BLOSUM62 scoring matrix for protein sequence insertion by TIPars.**
(XLSX)

**S7 Table. RF distances of IQTREE2, PAGAN2 and MAPLE for multiple taxa insertion.** RF distances of inserted trees against the reference tree were large and were not used for performance comparison.
(XLSX)

**S8 Table. RF distance per query taxon for every test set.** After adding all query taxa, for each query taxon the other queries were pruned from the result and reference trees, and their RF distance was computed. The RF distance per query taxon is the mean of over all query taxa in each test set. P-values were calculated by the right-sided paired t-test.
(XLSX)

**S9 Table. Cases of different Gamma20 log-likelihoods for trees resulting from TIPars and UShER.** For the random100, random1000, successive100 and successive1000 datasets, Gamma20 log-likelihoods were computed after optimization of tree branch lengths by FastTree2. "TIPars>UShER" indicates the loglikelihood of the tree resulting from TIPars is higher than that from UShER, and vice-versa.
(XLSX)

**S10 Table. Run time and memory usage to optimize the branch lengths of a SARS-CoV-2 100k taxa tree.** Tests were run 100 times using FastTree2 (double-precision version) on a

server with eight Intel Xeon Gold 6242R CPU cores.
(XLSX)

**S11 Table. Accuracy of PANGO lineages.** The statistic is the accuracy of the PANGO lineages for twenty sets of 100, 1000, 5000 and 10000 novel samples inserted into the SARS-CoV-2 100k taxa tree.
(XLSX)

**S12 Table. Number of cases for the resulting trees with both distal and sibling branch length smaller than 1E-8 in the newly added branch and the responding RF distance equal to 0 against the reference polytomous tree.**
(XLSX)

**S13 Table. Performance comparisons of single taxon placement and multiple taxa insertion.** For single taxon placement, the ranking is considering both placement accuracy and RF distance for the performances of TIPars, UShER and EPA-ng as presented in Fig 1. For example, in 16S dataset, though TIPars and EPA-ng achieved the same placement accuracy (81.3%), EPA-ng had a smaller mean RF distance (0.16) than TIPars (0.27) and would be ranked as better. For multiple taxa insertion, the ranking is considering both tree log-likelihood and RF distance for the performances of TIPars, UShER and IQTREE2 as presented in Fig 3. A caveat is that we have not compared IQTREE2 with RF distance because it may alter the input tree topology after inserting sequences, which would lead to a bias to its high tree log-likelihood. 'A > B' presents A is better than B.
(XLSX)

**S14 Table. Runtime and memory usage for 16S, H3N2 and NDV dataset preparation.** Tests were run on a server with 32 Intel Xeon Gold 6242 CPU cores.
(XLSX)

**S15 Table. Number of sequences for BA related PANGO Lineages in the newest SARS-CoV-2 sequences from January 1 to June 4, 2022.**
(XLSX)

**S16 Table. Commands to run the different tools that were tested.**
(XLSX)

## Acknowledgments

We gratefully acknowledge the following Authors from the Originating laboratories responsible for obtaining the specimens and the Submitting laboratories where genetic sequence data were generated and shared via GISAID Initiative, on which this research is based. A full acknowledgement table can be found with four EPI_SET-IDs, i.e., EPI_SET_20220531kz, EPI_SET_20211201vz, EPI_SET_20211206tc and EPI_SET_20220701rg, in Data Acknowledgement Locator under GISAID resources (https://www.gisaid.org/). We thank Vivian Leung for her edits and comments on the manuscript.

## Author Contributions

**Conceptualization:** Yongtao Ye, Marcus H. Shum, Guangchuang Yu, Tommy Tsan-Yuk Lam.

**Data curation:** Yongtao Ye, Marcus H. Shum, Joseph L. Tsui, Guangchuang Yu.

**Formal analysis:** Yongtao Ye.

**Funding acquisition:** Huachen Zhu, Joseph T. Wu, Yi Guan, Tommy Tsan-Yuk Lam.

**Investigation:** Yongtao Ye.

**Methodology:** Yongtao Ye.

**Project administration:** Tommy Tsan-Yuk Lam.

**Resources:** Yongtao Ye, Marcus H. Shum.

**Software:** Yongtao Ye, Guangchuang Yu, Tommy Tsan-Yuk Lam.

**Supervision:** Tommy Tsan-Yuk Lam.

**Validation:** Yongtao Ye.

**Visualization:** Yongtao Ye, Marcus H. Shum.

**Writing – original draft:** Yongtao Ye.

**Writing – review & editing:** Yongtao Ye, Marcus H. Shum, Joseph L. Tsui, David K. Smith, Huachen Zhu, Joseph T. Wu, Yi Guan, Tommy Tsan-Yuk Lam.

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
