## [Decision Letter · Decision Letter 0]

26 Oct 2023

Dear Dr. Lam,

Thank you very much for submitting your manuscript "Robust expansion of phylogeny for fast-growing genome sequence data" for consideration at PLOS Computational Biology.

As with all papers reviewed by the journal, your manuscript was reviewed by members of the editorial board and by several independent reviewers. In light of the reviews (below this email), we would like to invite the resubmission of a significantly-revised version that takes into account the reviewers' comments.

The reviewers all agreed that this study is meritorious, and I agree. Each reviewer raised important points that must be addressed in a revision. Further, I agree with two of the reviewers that a comparison against MAPLE would greatly benefit this manuscript.

We cannot make any decision about publication until we have seen the revised manuscript and your response to the reviewers' comments. Your revised manuscript is also likely to be sent to reviewers for further evaluation.

Sincerely,

Joel O. Wertheim

Academic Editor

PLOS Computational Biology

James O'Dwyer

Section Editor

PLOS Computational Biology

The reviewers all agreed that this study is meritorious, and I agree. Each reviewer raised important points that must be addressed in a revision. Further, I agree with two of the reviewers that a comparison against MAPLE would greatly benefit this manuscript.

Reviewer's Responses to Questions

**Comments to the Authors:**

Reviewer #1: This work (TIPars) integrates parsimony analysis with pre-computed ancestral sequences. It took only 21 seconds to insert 100 genomes into a 100k-taxa reference tree using an efficient 1.4 GB peak memory, outperforming other methods for moderately similar sequences (in terms of accuracy), although second to UShER for highly similar and divergent scenarios. This study is well designed and done in presenting detailed metrics of performance (RF distances, log-likelihood, running time, peak memory usage, etc.) for various datasets and scenarios, including leave-one-out insertion, leave-many-out insertion, and inserting novel sequences. Results are well discussed and interpreted. The manuscript is well written. I only have a number of minor suggestions below.

1. It was mentioned in line#98 that “RAPPAS and PAGAN2 were unable to complete within 96 hours, hence, no data were available.” In Table 1 caption, it was mentioned they “could not be run on this dataset”. It’s better to clarify that no success was due to that the job could not be done within 96 hours. Was it running but simply did not get result in time? Or there were factors preventing the program to run anyway?

2. It would probably be better that Fig 1B has its own chart legends as in Fig 1C and Fig 1D does, although they were texted in the figure legend. Similar concerns are also found in other figures.

3. As stated in line#318 to #323, TIPars could be compromised with incorrect placements accumulated as multiple sequences are inserted sequentially. The authors suggest conducting tree refinement for remedy. It will be great if one such example can be included in this manuscript, if possible.

Reviewer #2: In this manuscript, Ye at al. present TIPars, a phylogenetic placement algorithm for the rapid expansion of microbial phylogenies. In particular, according to the authors, TIPars achieves a better balance of efficiency and accuracy relative to prior tools and works well for similar as well as divergent sequences. Supporting evidence has been provided using single and multiple taxa insertion on four different datasets (SARS-CoV-2, Influenza, NDV and 16S). Source code is provided and is also packaged into a web app. While the manuscript is easy to read and aims to solve an important problem, there are certain aspects of the manuscript that need improvement, as I highlight below.

First, to my understanding, TIPars uses a very similar parsimony-based placement strategy as UShER. However, it differs in the reconstruction of ancestral sequences, using likelihood instead of parsimony. This seems to help it achieve a better trade-off of efficiency and accuracy. However, a more sophisticated approach, but in a similar vein, was recently developed in MAPLE (De Maio et al., Nature Genetics 2023). In particular, MAPLE uses a “Parsimonious Likelihood” approach that also achieves a good speed-accuracy trade-off, and therefore, a comparison to it is warranted.

Second, while the authors talk about TIPars’ application to phylogeny expansion, I don’t think this ability has been convincingly demonstrated. Importantly, greedy placement strategies are known to accumulate suboptimalities during phylogenetic expansion. This issue is ameliorated in SARS-CoV-2 UShER trees by using matOptimize (Ye et al., Bioinformatics 2022). MAPLE also uses SPR moves to optimize the topology periodically. Though the authors have experimented with multiple taxa insertion, the number of taxa inserted is small relative to the size of the starting phylogeny, and it is unclear if TIPars would be able to accurately maintain the expanding phylogeny. The scalability of TIPars’ approach would likely also be limited by tby the methods to build the MSA and the likelihood-based reference tree. I would like to see the authors address this important issue in their revision.

Lastly, the authors are probably not using the latest versions of the baseline tools. For example, they used v0.3.8 for UShER. UShER v0.6 provides additional optimizations and includes usher-sampled, which is over 10x faster while guaranteeing the same placements. It’s also not clear why TIPars is ~10x slower and ~2x less memory efficient than UShER though its placement strategy is similar. I hope the authors can expand on this.

A few minor suggestions and questions for the authors:

1. How did the authors obtain the PANGO lineage assignments for the SARS-CoV-2 sequences? To my understanding, the PANGO system has switched to UShER (called PUSHER) for lineage assignments.

2. What were the runtimes required to generate the MSAs, reference trees and the ancestral sequences for the different datasets? How frequently does this need to be done to maintain accuracy?

3. Is there a limit to the tree size that TIPars can handle? Why not evaluate the performance on millions of SARS-CoV-2 sequences that are available?

4. Does PastML use maximum parsimony (line 373)? I thought the goal was to infer ancestral sequences using likelihood.

5. Why have the authors used different methodologies for generating reference trees of different datasets (IQ-Tree GTR for one and RaxML GTG+G in the remaining)?

6. In addition to RF distance, it might be helpful to also evaluate the distance from the correct node.

7. Typo: “verified a superiority” -> “verify the superiority”

Reviewer #3: In their Manuscript Ye et al. describe a novel method for placing taxa on an existing phylogenetic tree. Similar 'online' phylogenetic approaches have proven very useful for analyses of large, growing datasets since they avoid the computational burden of reestimating large phylogenies from scratch each time new data becomes available. Existing approaches typically use maximum likelihood, minimum evolution or maximum parsimony to determine where a new sequence falls on an existing tree. As the the authors discuss, there is a trade-off between accuracy and speed, with full likelihood methods requiring longer run times to reach more accurate reconstructions. The proposed method, TIPars, provides a flexible compromise between the accuracy of maximin likelihood and the speed of parsimony.

TIPars evaluates taxa placement by estimating the number of substitutions between a query taxa and the precomputed ancestral sequence present on each branch. Because TIPars is agnostic to how these ancestral sequences are computed, the user is able fine tune the method to their needs. More accurate, slower results may be achieved if the ancestral states are estimated with maximum likelihood. However, this step is only needed once. Because placement is based on a simple parsimony-like score it remains efficient.

The author rigorously benchmark TIPars against standard tools with an array of datasets that vary in size and diversity. The tool is available as a command line interface as well as a well a web service, which should make it easy to use and share results. I have only the following concerns with the manuscript.

The authors show that TIPars outperforms USHER in a number of settings. One of these, is in SARS-CoV-2 datasets where both USHER and TIPars use parsimony reconstructions. It is unclear why TIPars does better, when similar methods are used. The authors suggest indels may be informing the taxa placement, but the methods don't mention how indels are used, and it seems like the MSA was generated by aligning to a reference.

Possibly related to the point above, it would be good of the authors to provide some of the reasoning behind their tie-breaking criteria. Are these criteria better suited for certain datasets like densely sampled outbreaks?

I hesitate to suggest more tools to benchmark against, but it would be informative to include the approximate likelihood used in MAPLE (De Maio, 2023). This approximation may break down in more divergent datasets, but has been shown to perform well for large, low-diversity datasets.

The authors should provide more discussion around how TIPars could be used in a large outbreak. The need to build and maintain a large MSA is more intensive than current approaches that align to a reference, and how could users apply the NNI and SPR moves suggested in line 323? (I believe USHER allows for these moves, but using USHER to do so would probably defeat the purpose).

In the PANGOlins analysis, how accurate are the pangolin lineages assigned to tips in the full tree? The accuracy of the lineage designations is assumed correct, but this might not be the case. PANGO-lineages have been defined using different trees overtime (first iqtree and later USHER).If the full iqtree built here is better than the one used during designation the correct placement may lead to the wrong pango lineage.

Methods 330: The wording here is a little unclear. It seems like the number of sites that differ between Q, A and P is summed not the number of mutations since the example given would require at least 2 mutations.

Line 342: What is sigma in the equation for l_{P-Q}?

**Have the authors made all data and (if applicable) computational code underlying the findings in their manuscript fully available?**

Reviewer #1: Yes

Reviewer #2: None

Reviewer #3: Yes

PLOS authors have the option to publish the peer review history of their article (what does this mean?). If published, this will include your full peer review and any attached files.

Reviewer #1: No

Reviewer #2: No

Reviewer #3: No
---

## [Decision Letter · Decision Letter 1]

8 Jan 2024

Dear Dr. Lam,

Thank you very much for submitting your manuscript "Robust expansion of phylogeny for fast-growing genome sequence data" for consideration at PLOS Computational Biology. Based on the second round of reviews, we are likely to accept this manuscript for publication, providing that you modify the manuscript according to the review recommendations.

Thank you for your extensive revisions in response to the Reviewers' comments and criticism, which have resulted in a substantially improved manuscript nearly ready for acceptance. However, one of the Reviewers (#2) still has a few queries that need to be addressed prior to acceptance. I would like to see one last version of this manuscript that addresses these issues. Note, I do not believe a full comparison to UShER-Sampled is needed to justify publication of a TIPars manuscript (point #2). That said, this manuscript would benefit from an explicit acknowledgement of these updated tool and a discussion of the implications on the findings presented.

Sincerely,

Joel O. Wertheim

Academic Editor

PLOS Computational Biology

James O'Dwyer

Section Editor

PLOS Computational Biology

Thank you for your extensive revisions in response to the Reviewers' comments and criticism, which have resulted in a substantially improved manuscript nearly ready for acceptance. However, one of the Reviewers still has a few queries that need to be addressed prior to acceptance. I would like to see one last version of this manuscript that addresses these issues. Note, I do not believe a full comparison to UShER-Sampled is needed to justify publication of a TIPars manuscript. That said, this manuscript would still benefit from an explicit acknowledgement of these updated tool and its implications on the findings presented.

Reviewer's Responses to Questions

**Comments to the Authors:**

Reviewer #1: The authors have addressed all questions well, and appropriately updated related texts and figures.

Reviewer #2: The revised manuscript shows significant improvement and the authors have successfully addressed most of my previous concerns. However, there are still a few unresolved issues and therefore, I recommend a minor revision.

1. The authors are incorrect in stating that UShER and MAPLE also require an MSA for phylogeny expansion as those tools are reference-based, and therefore, they only require pairwise alignments of new sequences to a single reference sequence. Because inferring an MSA is significantly more expensive than inferring a set of pairwise alignments to a reference, I feel this is a serious limitation of TlPars currently.

2. The authors did not incorporate my suggestion to evaluate usher-sampled. This is a new program with several optimizations which have been recently described in Hinrichs et al. Nature Genetics 2023.

3. The authors did not satisfactorily answer my question about the scalability of TlPars. Even though a fair comparison with millions of SARS-CoV-2 sequences may not be possible, it would be helpful to understand if TlPars can handle those many sequences.

4. In Fig. 4C, can the authors explain why the parsimony score is decreasing when more sequences are being added? I think this should not happen.

Reviewer #3: I appreciate the time and effort the authors have taken to address mine and the other reviewers' comments. I have no remaining concerns with the manuscript.

**Have the authors made all data and (if applicable) computational code underlying the findings in their manuscript fully available?**

Reviewer #1: Yes

Reviewer #2: Yes

Reviewer #3: Yes

PLOS authors have the option to publish the peer review history of their article (what does this mean?). If published, this will include your full peer review and any attached files.

Reviewer #1: No

Reviewer #2: No

Reviewer #3: No

Figure Files:

Data Requirements:

Reproducibility:

References:

---

## [Editor Report · Decision Letter 2]

29 Jan 2024

Dear Dr. Lam,

We are pleased to inform you that your manuscript 'Robust expansion of phylogeny for fast-growing genome sequence data' has been provisionally accepted for publication in PLOS Computational Biology.

Best regards,

Joel O. Wertheim

Academic Editor

PLOS Computational Biology

James O'Dwyer

Section Editor

PLOS Computational Biology

---

## [Editor Report · Acceptance letter]

4 Feb 2024

PCOMPBIOL-D-23-01415R2 

Robust expansion of phylogeny for fast-growing genome sequence data

Dear Dr Lam,

I am pleased to inform you that your manuscript has been formally accepted for publication in PLOS Computational Biology. Your manuscript is now with our production department and you will be notified of the publication date in due course.

With kind regards,

Zsofia Freund
